# A Novel TLR4-SYK Interaction Axis Plays an Essential Role in the Innate Immunity Response in Bovine Mammary Epithelial Cells

**DOI:** 10.3390/biomedicines11010097

**Published:** 2022-12-30

**Authors:** Fan Yang, Lu Yuan, Minghui Xiang, Qiang Jiang, Manling Zhang, Fanghui Chen, Jie Tong, Jinming Huang, Yafei Cai

**Affiliations:** 1College of Animal Science and Technology, Nanjing Agricultural University, Nanjing 210095, China; 2Department of Human Anatomy, Bengbu Medical College, Bengbu 233030, China; 3Key Laboratory of Cardiovascular and Cerebrovascular Diseases, Bengbu Medical College, Bengbu 233030, China; 4Institute of Animal Science and Veterinary Medicine, Shandong Academy of Agricultural Sciences, Jinan 250100, China

**Keywords:** TLR4–SYK interaction axis, mastitis, bMECs, dairy cattle

## Abstract

Mammary gland epithelium, as the first line of defense for bovine mammary gland immunity, is crucial in the process of mammary glands’ innate immunity, especially that of bovine mammary epithelial cells (bMECs). Our previous studies successfully marked SYK as an important candidate gene for mastitis traits via GWAS and preliminarily confirmed that SYK expression is down-regulated in bMECs with LPS (*E. coli*) stimulation, but its work mechanism is still unclear. In this study, for the first time, in vivo, TLR4 and SYK were colocalized and had a high correlation in mastitis mammary epithelium; protein–protein interaction results also confirmed that there was a direct interaction between them in mastitis tissue, suggesting that SYK participates in the immune regulation of the TLR4 cascade for bovine mastitis. In vitro, TLR4 also interacts with SYK in LPS (*E. coli*)-stimulated or GBS (*S. agalactiae*)-infected bMECs, respectively. Moreover, TLR4 mRNA expression and protein levels were little affected in bMECs^SYK-^ with LPS stimulation or GBS infection, indicating that SYK is an important downstream element of the TLR4 cascade in bMECs. Interestingly, *IL-1β*, *IL-8*, *NF-κB* and *NLRP3* expression in LPS-stimulated or GBS-infected bMECs^SYK-^ were significantly higher than in the control group, while *AKT1* expression was down-regulated, implying that SYK could inhibit the *IL-1β*, *IL-8*, *NF-κB* and *NLRP3* expression and alleviate inflammation in bMECs with LPS and GBS. Taken together, our solid evidence supports that TLR4/SYK/NF-κB signal axis in bMECs regulates the innate immunity response to LPS or GBS.

## 1. Introduction

Mastitis is the most challenging mammary gland inflammatory disease of dairy cows, and it seriously affects the cows’ health and profitability of dairy pastures [1,2]. Invasion and colonization of pathogenic bacteria in breast tissue is the main cause of mastitis [3,4], and the clinical symptoms of mastitis are pathogen-specific [5,6]. The ability to resist pathogen infection within the mammary gland depends on the efficiency of the mammary gland immune system, which not only prevents the pathogens from invading the mammary glands but also effectively eliminates pathogen infections and promotes the restoration of normal functions of mammary gland tissues [7,8,9]. The innate and adaptive immune systems composed of complex components of tissues, cells and molecules work together to provide the optimal protection for breast health [10,11]. In particular, the first line of defense represented by mammary epithelium is to effectively initiate the immune response, killing and eliminating pathogens before any abnormal changes in breast tissue occur, which is crucial for the resistance of and impacts susceptibility to breast infection to mastitis [9,12,13]. Once the pathogen successfully breaks through the physical defenses of the nipple end, mammary cell populations (lymphocytes, mammary epithelial cells, endothelial cells, etc.) can initiate intracellular immune-related signal axes via specific pattern recognition receptors [14,15,16,17], such as TLRs/NF-κB, LPS/CD14/TNF-α, JAK-STAT, etc., stimulate immune-related cells to release proinflammatory factors (TNF-α, IL-1β, IL-8, etc.) and recruit neutrophils and macrophages into mammary endophagtic pathogens. Here, mammary epithelial cells (MECs) play an important role and are irreplaceable. Furthermore, MECs initiate different signal cascades for response to different pathogen infections [18,19], for instance, activating the TLR2/TLR4-NF-κB pathway for response to *E. coli* infection, while activating the TLR2/IL17A pathway response to *S. aureus* and the TLR2/TLR4-IL-1β/IL-6 pathway response to mycoplasma. TLR2 and TLR4 are particularly important in mammary gland immune defense because they are pathogen-associated molecular pattern ligands (PAMPs) that cause mastitis linked to Gram-positive (peptidoglycan, LTA) and Gram-negative (lipopolysaccharide, LPS) pathogens, including *S. aureus*, *S. uberis* and *E. coli* [14,16,20]. Moreover, in the early or late stages of breast infection with pathogens, MECs and their recruited immune cells can secrete specific innate immunosoluble cytokines or chemokines (CCL2, CCL 5, CCL 20, CXCL10, TNF-α, MyD88, etc.) [16,18,21,22]. Therefore, it is not only a huge challenge to study the immunoregulatory strategies of MECs to different pathogens but also an interesting and meaningful issue that attracts the eyes of many researchers.

Spleen tyrosine kinase (SYK) is a cytoplasmic protein nonreceptor tyrosine kinase that plays a vital role in immune cell response (adaptive immunity and innate immunity) and nonimmune cell and other unexpected biological functions, especially in mediating those including innate immune recognition, cell adhesion, brown fat formation, blood vessel development, osteoclast maturation and platelet activation [23,24,25,26]. In hematopoietic-derived cells, SYK is required for Fcgamma receptor signaling in macrophages and neutrophils [27] and for BCR activation in B cells [28], while it coupling with IL-1 secretion sensitizes allergy to hapten and proinflammatory responses in dendritic cells [29]. Moreover, SYK has a matching unique expression pattern with the transcription factor KLF5, ZNF608 and c-MAF in human basophils [30], while it activates the NLRP3 inflammasome in human monocytes [31] and unifies human platelet activation [32]. In nonhematopoietic cells, SYK mediates the differentiation and activation of brown fat and is an important mediator of brown adipogenesis and function [24]. SYK is also a potentially effective cell migration inhibitor in epithelia, including murine/rat mammary epithelium, skin epithelium, and myoepithelium, but does not have an effect in lung and intestinal epithelia [33]. Furthermore, SYK phosphorylation affects the inflammatory response of gastric mucosal epithelium to *H. pylori* and mediates high glucose signal transduction in proximal tubule epithelial cells with TLR4 [34]. Moreover, SYK regulates the proliferation of bovine MECs and affects mammary gland remodeling [35]. However, the TLR4/NF-κB pathway plays a key role in bovine MECs inflammation induced by LPS (*E. coli*) [20]. SYK coupled with the cytoplasmic domain of TLR4 and plays an important role in the TLR4 cascade response in dendritic cells [36], and in neutrophils and monocytes as well [37], but little is known in MECs, especially in bovine MECs.

To investigate the correlation of SYK and TLR4 in the innate immunity response of bMECs, this study focuses on the differential expression of SYK and TLR4 before or after LPS stimulation or *S. agalactiae* (GBS) in bovine MECs via siRNA, qRT-PCR, Western blot (WB) and immunofluorescence. We preliminarily explore the position of the TLR4–SYK signaling axis in bMECs with LPS or GBS. Our research provides a research basis for understanding the defense and immune response of bovine mammary glands against pathogenic infections and new strategies for the prevention and treatment of mastitis.

## 2. Materials and Methods

### 2.1. Experimental Animals

The animal-related procedures were approved by the Animal Welfare Committee of Nanjing Agricultural University, China, with the approval number 20160615. All animal experiments were performed in strict accordance with the guidelines and rules set by the committee.

Mammary gland tissue samples were obtained from six Chinese Holstein cows for experimental research. Sample selection standard: from the same pasture, with the same nutritional standards and growing environment, 3rd early lactation (15–60 days). The samples were divided into two groups: normal health group (3 cows, mean of somatic cell count (SCC) value = (18.7 ± 3.1) × 10^4^ < 30 × 10^4^ cells/mL) and clinical mastitis group (3 cows, mean of SCC value = (1558.5 ± 101.9) × 10^4^ > 100 × 10^4^ cells/mL, mammary gland tissue: red, swollen, fever, painful, etc.). Breast lobular tissue (approximately 1–2 cm^3^ in size, avoid connective tissue and adipose tissue as much as possible) from dairy cows was removed via biopsy, placed in 0 °C DMEM/F12 complete culture medium and quickly brought back to the sterile laboratory. We then prepared for the next experiment, such as total RNA and protein extractions, isolation and primary culture of bovine mammary epithelial cells (bMECs).

### 2.2. Total Tissue RNA Extraction and mRNA Expression Analysis

Total RNA was isolated from mammary gland tissue using TRIzol^TM^ Reagent (Cat#15596-026, ThermoFisher, WaItham, MA, USA), and RNA concentration was quantified by the Spark^TM^ multimode microplate reader. First-strand cDNA was synthesized by incubating 1 µg of total RNA with RevertAid first-strand cDNA synthesis kit (#K1621, Thermo Scientific with gDNA Eraser, WaItham, MA, USA), according to the manufacturer’s instructions. Quantitative PCR was performed on a real-time PCR system, including Bio-RadCFX Manager^TM^ (Åercules, CA, USA) and SuperReal (SYBR Green, Bio-Rad, Åercules, CA, USA) fluorescence quantification kit (#FP205, TIANGEN Biotech, Beijing, China). The primers and procedures are listed in Appendix A. All experimental operations follow the MIQE guideline [38,39].

### 2.3. Tissue Total Protein Extraction and Protein Level Analysis

The traditional RIPA lysis solution method (including PMSF) was used to extract the total protein of mammary gland tissue and bMECs (Cat#P0013C, Beyotime Biotechnology, Shanghai, China), and the total protein concentration was determined by the BCA method protein assay kit (Cat#P0012, Beyotime Biotechnology, Shanghai, China). Protein levels of SYK and related proteins were analyzed by conventional WB techniques. Antibodies for this study include Anti-SYK (Cat#sc-166226, Santa Cruz Biotechnology, Dallas, Texas, USA), Anti-SYK (Cat#ab40781, Abcam, Cambridge, UK), Anti-pSYK (p-Y319.17A, Cat#sc136248, Santa Cruz Biotechnology, Dallas, Texas, USA), Anti-TLR4 (Cat#sc293072, Santa Cruz Biotechnology, Dallas, Texas, USA), Anti-NLRP3 (Cat#15101, Cell Signaling Technology, USA), Anti-IL-1β (Cat#12703, Cell Signaling Technology, Beverly, MA, USA), AffiniPure Goat Anti-Mouse IgG (Cat#BA1038, BOSTER, Wuhan, China) and AffiniPure Goat Anti-Rabbit IgG (Cat#BA1039, BOSTER, Wuhan, China). The proteins were boiled in SDS-PAGE sample buffer (62.5 mM Tris-HCl, pH 6.8; 2% SDS; 10% glycerol; and 0.1% bromophenol blue) containing β-ME, analyzed by SDS-PAGE and transferred onto PVDF (Bio-Rad) membranes, which were subsequently blocked in 5% nonfat milk (wt./vol in PBS) for 1 h at room temperature and immunoblotted with the indicated primary antibodies at 4 °C overnight and appropriate horseradish peroxidase (HRP)-conjugated secondary antibodies for 1 h at room temperature, washed and then probed.

### 2.4. Isolation, Culture and Identification of Pathogenic Microorganisms in Milk

Isolation of pathogenic bacteria: Under aseptic conditions, pathogenic bacteria in the milk sample were isolated and purified using five selective mediums (MacConkey Agar) (MAC, isolation of Gram-negative bacteria), Triple Sugar Iron Agar (TSI, isolation of *Escherichia coli*, *Salmonella*, *Shigella*), *Salmonella-Shigella* Agar (SS, isolation of *Salmonella* and *Shigella*), Gauze’s Synthetic Medium No. 1 (GSMN1, isolation of actinomycetes) and Columbia Blood Agar (CBA, isolation of hemolytic streptococcus) in combination with plate scribing (four-zone scribing method): Milk samples were dipped in inoculation rings and inoculated into the above five media to isolate and purify pathogenic bacteria in the milk samples. We inverted the culture dish and placed it in a 37 °C incubator for 24 h to obtain a single colony clone of the pathogens; we repeated this once. Then, the single colony of the pathogen was placed in the broth and cultured at 37 °C for 24 h (220 rpm/min), and we obtained a single pathogen flora. Identification of pathogenic bacteria: Gram stain was used to identify pathogenic bacteria in milk samples that were negative or positive. Pathogen bacteria genomic DNA was extracted and purified using Rapid Bacterial Genome Isolation Kit (#B518225, Sangon Biotech, Shanghai, China). PCR reaction system and procedure: The total PCR volume was 50 μL, including 5 μL 10× Buffer (with Mg^2+^), 1 μL dNTPs, 2 μL upstream primer [LPW57 5′-AGTTTGATCCTGGCTCAG-3′ (NO. 10th–27th bp)] and 2 μL downstream primer [LPW58 5′-AGGCCCGGGAACGTATTCAC-3′ (NO. 1370th–1389th bp)]. Then PCR products were detected by 2% agarose gel electrophoresis and identified by 16S ribosomal RNA (16S rRNA) gene sequencing (Appendix A, NCBI database). Streptococcus agalactiae (GBS) is one of the pathogenic microorganisms.

### 2.5. Isolation, Identification and Primary Culture of bMECs In Vitro

Isolation of bMECs: First, we chose healthy cows [40]. Then bMECs were isolated by the tissue block inverted adherent method [41,42], and the specific operation process refers to our previous research [43]. Identification of bMECs: bMECs were identified by immunofluorescence staining through epithelial sensitivity to cell membrane Keratin-18 (C-18, Cat#sc-51582, Santa Cruz, Dallas, TX, USA). Primary culture of bMECs: 1 × 10^5^ cells/mL cell suspension, seeded on a 6-well culture plate, placed in a 37 ℃, 5% CO_2_ cell incubator; we replaced the DMEM/F12 complete culture medium every 48 h for the subculture and within 20 generations for the primary culture (Appendix A).

Climbing sheets of bMECs: We put the sterile antistripping slide into a sterile petri dish (labeled), took 100 μL of 1 × 10^5^ cells/mL cell suspension, inoculated it on the slide glass under aseptic conditions and then inoculated it at 37 ℃, in a 5% CO_2_ cell incubator, cultivated it for 24 h and changed the cell culture medium. Direct adhesion rate of cells with an inverted microscope was about 90%, indicating that the cell climb was basically complete. Fixation of bMECs: The cells that completed the climb were gently washed twice with 1× PBS; we absorbed the excess 1 PBS with absorbent paper and immediately fixed the slide cells with an appropriate amount (100 μL) of 4% paraformaldehyde, which stood at room temperature for about 20 min, and then sucked the 4% paraformaldehyde residue with absorbent paper, dried the result and directly performed subsequent immunofluorescence staining experiments or stored the slides at −20 ℃.

### 2.6. Lipopolysaccharide (LPS from E. coli) Stimulation Assay on bMECs In Vitro

bMECs with a concentration of 5 × 10^5^ cells/mL were passaged in 6-well plates for 24 h and stimulated with LPS (Cat#D8437, Sigma–Aldrich, St. Louis, MO, USA) derived from *E. coli*. The LPS concentration was 25 μg/mL, and the stimulation time program was referred to our previous research [43]. Total RNA and protein were extracted via All-In-One DNA/RNA/protein Mini-Preps Kit (#BS88003, Sangon Biotech, Shanghai, China) and stored at −80 ℃ for qRT-PCR and Western blot analysis.

### 2.7. Pathogenic Microorganism-Infected bMECs for In Vitro Validation

In vitro validation was performed in bMECs isolated from mammary gland tissues of three healthy dairy cows. Stimulating bMECs experiments followed Grunert and Stevens with a little change: bEMCs (5 × 10^5^) were placed in 6-well plates for 24 h and then stimulated with the pathogenic microorganisms multiplicity of infection (MOI = 10, pfu number per cell) for 60 min [15,44]. The total RNA of two group cells (stimulated and unstimulated) were extracted using the Trizol method and stored at −80 °C. The RT-qPCR reaction system and conditions were identical to the in vivo validation experiments.

### 2.8. SYK Gene siRNA in bMECs

Designing three pairs of siRNA sequences induced post-transcriptional silencing of the SYK gene in bMECs (Appendix A). Then SYK siRNA sequences were transfected into bMECs by LipoFiter^TM^ Liposomal Transfection Reagent (Cat#HB-TRLF-1000, HANBIO, Shanghai, China). The specific operation steps refer to the instructions of the transfection reagent. Cell concentrations used for the transfection experiments were 5 × 10^5^ cells/mL.

### 2.9. Immunofluorescence Colocalization for TLR4 and SYK in Bovine Mammary Gland Tissue and bMCEs

Bovine mammary gland tissue blocks (approximately 2 mm^3^) were made into paraffin sections using conventional paraffin slice techniques. The morphological changes between normal bovine mammary gland tissue and mastitis tissue were evaluated by conventional H&E staining. The correlation between SYK and TLR4 was analyzed in bovine mammary gland tissue via dual fluorescence staining: Permeabilized sections were blocked using 5% BSA for 1 h; *SYK* expression was significantly down-regulated in both mammary gland tissue Fluorescent antibodies: CoraLite488-conjugated Goat Anti-Mouse IgG (Cat#SA00013-1, Proteintech Group, Wuhan, China), Cy3-conjugated Affinipure Goat Anti-Rabbit IgG (Cat#SA00009-2, Proteintech Group, Wuhan, China).

### 2.10. Coimmunoprecipitation (Co-IP) for TLR4 and SYK in Bovine Mammary Gland Tissue and bMCEs

Bovine mammary gland tissue was lysed (operated on ice) by conventional RIPA lysate (including PMSF), and total protein concentration was detected by BCA protein assay kit, adjusted to 1 mg/mL, divided into 200 μL/clade and stored at −80 ℃ for reserve. The specific operation steps were as follows Protein G Agarose beads were prepared according to instructions (Cat#37478S, Cell Signaling Technology, Beverly, MA, USA) and incubated with 3 μg of the mouse or rabbit monoclonal antibodies overnight at 4 °C. The related beads were washed six times with IP buffer and boiled in SDS-PAGE buffer for 10 min at 95 °C. The samples were separated by SDS-PAGE gel and transferred onto PVDF membranes. Western blot experiments were performed with the indicated antibodies and visualized with Super-Signal West Pico chemiluminescent substrate (Pierce Chemical, Dallas, TX, USA). Then, the interaction between SYK and TLR4 was assessed in mammary gland tissue or bMECs via conventional WB.

### 2.11. Statistical Analysis Method

Statistical analysis of data was performed by Graph-Pad Prism 6 software; qRT-PCR results of differential gene mRNA expressions were analyzed by the relative quantitative method (2^−△△Ct^). ImageJ software was used according to the light of the target band in the WB image, and density values were converted into specific values to compare protein level differences. All data were analyzed via Student’s *t*-test and one-way ANOVA. Immunofluorescence colocalization analysis were also performed using ImageJ software (parameter standard: Pearson’s correlation coefficient, PCC).

## 3. Results

### 3.1. SYK Plays an Essential in Immune Response for Bovine Mastitis

Our previous GWAS study showed that SYK is an important candidate gene for bovine mastitis. GO classification and enrichment analysis results confirmed that SYK is enriched upon entry into ATP binding (molecular function), plasma membrane and cytosol (biological processes), and intracellular signal transduction (cellular components) (Figure 1), respectively. Protein–protein interaction (PPI, STRING version 11.5) analysis revealed that SYK is associated with TLR4 and NLRP3 (local clustering coefficient = 0.72, PPI enrichment *p*-value = 0.025 < 0.05, Appendix A). In addition, Encyclopedia of Genes and Genome (KEGG) analysis results also showed that SYK is involved in NF-κB, PI3K-AKT, B cell receptor, C-type lectin receptor and natural killer cell-mediated cytotoxicity signaling pathways (Appendix A). Public datasets for this research are available in the NCBI SRA database: PRJNA556499.

### 3.2. SYK, TLR4, NLRP3 and IL-1β Gene Expressions in Mammary Gland Tissue of Chinese Holstein

H&E staining revealed that the tissue framework in mastitis tissue was severely damaged: the normal structure of mammary gland was destroyed, including alveolar shrinkage and inflammatory cell invasion, and especially devastating was the destruction of the integrity of the alveolus structure (Figure 2A). Differential expression of SYK and related genes in bovine mammary tissues (normal and mastitis tissues) were detected via qRT-PCR and showed that *SYK* gene expression was significantly down-regulated in mastitis, while *AKT1*, *TLR4*, *TLR2*, *IL1β*, *IL-8*, *IL-10*, *NF-κB* and *NLRP3* gene expression were significantly up-regulated (Appendix A). Moreover, SYK protein level was also significantly down-regulated in bovine mastitis tissue, and protein levels of NLRP3, TLR4 and IL-1β were significantly up-regulated (Figure 2B,C). SYK phosphorylation level was also tested. Interestingly, p-SYK level decreased slightly, although there was no significant difference compared with SYK level in bovine mastitis tissue. The data also showed that SYK gene expression and protein levels have a strong consistency in bovine mastitis, and NLRP3, TLR4 and IL-1β as well. Regrettably, we did not find suitable antibodies to detect the protein levels for bovine TLR2, IL-8, IL-10 and NF-κB.

### 3.3. Colocalization of SYK and TLR4 in Bovine Mammary, Mastitis Tissues and bMECs

Our immunofluorescence colocalization revealed that TLR4 is predominantly localized in the cell membrane of bMECs, while SYK protein is mainly located in the cellular matrix of bMECs (Figure 3A). The Pearson’s correlation coefficient (PCC) of the immunofluorescence colocalization results of the above two proteins in the microenvironment of normal dairy cow mammary tissues was 0.5187 (0.4 < PCC < 0.6), which means a low-to-moderate correlation. Furthermore, the PCC value of immunofluorescence colocalization of TLR4 and SYK in bovine mastitis tissue was 0.8857 (0.6 < PCC < 1.0), which was highly correlated. Figure 4 shows the results of immunofluorescence colocalization between TLR4 and SYK in bMCEs: the PCC was 0.6277 in control group, and the PCC was 0.8667 (30 min) and 0.8546 (60 min) in bMCEs with LPS, while it was 0.8912 (30 min) and 0.8611 (60 min) in bMCEs with GBS (Appendix A). The coimmunoprecipitation analysis also confirmed the direct interaction of TLR4 and SYK in bovine mammary and mastitis tissues (Figure 3B,C).

### 3.4. SYK Protein Levels Showed Time-Course Effect in bMECs with LPS Stimulation or GBS Infection

We focused on the levels of SYK protein in bMECs with LPS stimulation or GBS infection. Data revealed that SYK protein levels were down-regulated in bMECs with stimulation, of which significantly down-regulated at 30 min (Figure 5A) after LPS stimulation. Although with the prolongation of the stimulation time the protein level began to turn back after 30 min, it was still down-regulated until the end of the stimulation time (120 min), and there was a significant difference (*p* < 0.05). Interestingly, similar phenomena also appeared in GBS-infected bMECs, but what puzzled us was that SYK levels showed a downward trend during 60–120 min (Figure 5B). Further, the correlation between TLR4 and SYK in bMECs with LPS stimulation or GBS infection by using coimmunoprecipitation confirmed that there was direct interaction between TLR4 and SYK in bMECs (Figure 5C,D).

### 3.5. SYK, TLR4, NLRP3 and IL-1β Expressions in bMECs and SYK siRNA bMECs (bMECs^SYK-^) with LPS Stimulation or GBS Infection

We designed three SYK gene siRNA interference sequences, among which siRNA1 sequence (*p* < 0.01) had the best interference effect (Figure 6A). *SYK* gene expression was significantly down-regulated in bMECs^SYK-^ with LPS stimulation for 30 min; the RT-PCR suggested that the interference effect of siRNA is significant (Appendix A), while there was no difference in bMECs^SYK-^ with GBS stimulation infection. The mRNA expression of *TLR2*, *TLR4*, *NLRP3*, *AKT1*, inflammatory factors (*IL-1β*, *IL-8*, and *IL-10*) and transcription factor (NF-κB) were also detected, and we found that there were no differences in mRNA expression between bMECs and bMECs^SYK-^, implying that bMECs^SYK-^ is still able to complete normal cellular functional activity (Figure 6B–D). In bMECs and bMECsSYK- stimulated by GBS and LPS, the expressions of TLR2, TLR4 (*p* < 0.05), IL-8 (*p* < 0.001), IL-10 (*p* < 0.01), AKT1 (*p* < 0.01) and NLRP3 (*p* < 0.001) were up-regulated in bMECs, while IL-1β and NF-κB down-regulated in LPS-stimulated bMECs. Moreover, under LPS and GBS stimulation, IL-1β (*p* < 0.01), IL-8 (*p* < 0.001), IL-10 (*p* < 0.01), NF-κB (*p* < 0.01) and NLRP3 (*p* < 0.001) were significantly up-regulated, while AKT1 markedly down-regulated (*p* < 0.01).

### 3.6. SYK, TLR4, NLRP3 and IL-1β Protein Levels in bMECs and bMECs^SYK-^ with LPS Stimulation or GBS Infection

In bMECs^SYK-^, the SYK level was significantly down-regulated (*p* < 0.01), and the phosphorylation level of *p*-SYK (Tyr319 and Tyr352) was also significantly down-regulated (*p* < 0.01). Data also revealed that there was no significant difference between SYK and p-SYK, namely, the SYK protein had high phosphorylation at Tyr319 and Tyr352, suggesting that SYK was in an activated state in bMECs (Figure 7). The TLR4 level was significantly up-regulated in both bMECs and bMECs^SYK-^ with LPS stimulation or GBS infection, suggesting that TLR4 plays an important in the immune response of bMECs. Furthermore, the NLRP3 level (*p* < 0.05) was significantly up-regulated in bMECs with LPS, while it was even higher up-regulated in bMECs^SYK-^. However, interestingly, the NLRP3 level changed little in GBS-infected bMECs, while it was significantly up-regulated in bMECs^SYK-^, indicating that SYK can inhibit the NLRP3 level in bMECs. Moreover, IL-1β levels were significantly up-regulated in LPS-stimulated or GBS-infected bMECs and bMECs^SYK-^, and the up-regulation was higher with LPS stimulation, with the highest being found in bMECs^SYK-^.

## 4. Discussion

### 4.1. SYK Plays a Crucial Role in the Innate Immune Response of Bovine Mammary Gland Tissue and bMECs

Genome-wide association analysis (GWAS) provides appropriate research and control strategies for a better understanding of the genetic and biological basis of factors such as complex traits, diseases, etc., and bovine mastitis as well [45,46]. In this study, GO functional annotation and enrichment analysis results revealed that SYK was an important candidate gene for mastitis traits in Chinese Holstein. At the mRNA level, SYK expression was significantly down-regulated in both bovine mammary gland tissues and showed the same expression trend in LPS-stimulated and GBS-infected bMECs in vitro. More importantly, the SYK protein levels trend was similar to gene expression; that is, the SYK protein level was significantly down-regulated in bovine mastitis tissue and LPS-stimulated or GBS-infected bMECs, suggesting SYK plays a critical role in the immune response of bovine mammary gland tissue and bMECs in response to pathogenic bacterial infection. However, SYK phosphorylation represents the activation of SYK. Our research shows that the SYK total protein down-regulated but SYK activity was unchanged.

### 4.2. The SYK mRNA Expression and Protein Level in bMECs Responded to LPS Stimulation or GBS Infection Presented Time-Dependent Properties

Once the mammary gland is infected with foreign pathogenic bacteria, it will cause the mammary gland tissue to initiate the innate immune response [7,10]. Then mammary epithelial cells (MECs), as the first line of defense, will also promptly secrete multiple cytokines (such as IL-1β, IL-8 and TNF-α, etc.) and recruit a large number of lymphocytes (such as neutrophils) into the breast tissue to kill pathogens [22,47], resulting in a sharp increase in somatic cell count (SCC) of the milk in early and late breast infection and the corresponding mastitis phenotype. bMECs are the key to the rapid elimination of bacteria and prevention of mastitis [48,49]. In this experiment, LPS and GBS were used to stimulate and infect bMECs in vitro, respectively, and *SYK* expression was affected by LPS stimulation or GBS infection time, suggesting that the activity of exogenously stimulated bMECs is time-dependent [50,51]. In addition, genes such as TLR2, TLR4, NLRP3, IL-1β, IL-8, IL-10, AKT1 and NF-κB showed significant differences in mRNA expression before and after LPS stimulation or GBS infection, indicating their involvement in the immune response to bMECs. SYK protein levels were also affected by LPS stimulation or GBS infection time, which once again proves the results of the LPS stimulation or GBS infection time program research and the reliability of the in vitro validation model.

### 4.3. The TLR4–SYK Signaling Pathway Participates in the Immune Response of Bovine Mammary Gland Tissue and bMECs for Mastitis

Understanding the potential molecular mechanisms and immune regulation strategies of the susceptibility or resistance of dairy cow mammary gland tissues and mammary epithelial cells to pathogenic bacteria will be beneficial to the prevention and treatment of mastitis [19,22,52,53]. Our data revealed that the mRNA expression (*p* < 0.01) and protein levels (*p* < 0.01) of TLR2 and TLR4 were significantly up-regulated in bovine mastitis mammary gland tissues, suggesting that bovine mammary gland tissue initiates different innate immune regulatory pathways to respond to pathogenic infections [22,54]. In vitro, we stimulated or infected bMECs (including bMECs^SYK-^and bMECs) by using LPS and GBS, respectively, and found that the mRNA expression and protein levels did not differ in the two cells above, similar to the predecessors who infected bMECs through *E. coli* and *S. aureus* activated TLR2 and TLR4 [14]. In addition, our data also showed that the TLR2 cascade is mainly involved in the immune regulation of GBS-infected bMECs, while the TLR4 cascade is involved in the immune regulation of bMECs caused by LPS and GBS, which might be due to specific pathogen antigens initiating different immune responses in bMECs [18]. Previous studies have confirmed that the TLR4–SYK signal axis plays a key role in the immune response to LPS stimulation in neutrophils and monocytes [37]. Furthermore, SYK was down-regulated in peripheral blood neutrophils of dairy cows with mastitis (*E. coli*), participates in the TLR4 cascade, regulates the production of reactive oxygen species (ROS) in bovine neutrophils and affects neutrophils to kill pathogens [15,55,56]. Immunofluorescence colocalization data showed that TLR4 and SYK had a high correlation in bovine mastitis mammary gland tissue and bMCEs. In addition, coimmunoprecipitation results confirmed the presence of the TLR4–SYK signaling pathway in bovine mammary gland tissues and bMCEs to participate in the immune cascade response to mastitis and the same phenomenon in bMECs with LPS or GBS. Data also revealed that, under the same pathogenic factor, the expression of TLR2 and TLR4 showed little difference in bMECs and bMECs^SYK-^. In summary, SYK is a downstream ligand of the TLR2/4 [32,37], suggesting that TLR4–SYK is essential in the immune response of LPS stimulation or GBS infection of bMECs.

### 4.4. SYK Affected AKT1, IL-1β, IL-8, IL-10, NF-κB and NLRP3 Expression in bMECs with LPS or GBS

As a key player in the JAK2/STAT5 pathway, AKT1 plays an important role in bMECs differentiation, secretion, survival and proliferation, as well as in the regulation of breast remodeling and lactation sustainability [17,57,58]. JAk2 and STAT5, two key players in the immune cascade response, are associated with bovine mastitis susceptibility [59]. *AKT1* was significantly up-regulated in bovine mammary gland tissue in LPS-stimulated or GBS-infected bMECs, suggesting that there might be a SYK–AKT1 signal axis in bMECs to respond to LPS or GBS. Interestingly, compared with bMECs, it was significantly down-regulated in LPS-stimulated or GBS-infected bMECs^SYK-^, while the expression of *IL-1β*, *IL-8*, *IL-10*, *NF-κB* and *NLRP3* in bMECs^SYK-^ was higher than in bMECs, and NPRP3 protein level as well, indicating that these above genes were involved in the SYK-mediated immune response in bMECs. Therefore, based on the evidence from the experimental data, we propose the hypothesis that the potential TLR4/SYK/NF-κB/NLRP3 signal axis exists in bMECs to modulate the immune and inflammatory responses, although more research data are needed.

According to our research, the whole system is shown in Figure 8. When bMECs are stimulated by bacteria, the TLR4/SYK axis is implied. The expression of SYK and P-SYK markedly down-regulate. Then SYK reduction leads to the decrease in AKT1 expression and the increase in NF-κB expression. Finally, NF-κB enters nuclear to regulate IL-1β, il-8 and IL-10 transcriptions to trigger the whole inflammatory immunity response. NF-κB is positively related to the degree of inflammation of clinical acute mastitis, and upstream of NF-κB leads rapid reduction in milk level [60,61]. Previous studies have shown that alpinetin, cepharanie, curcumin, astragalin and thymol can release mastitis level via reducing TLR4, IκBα, NF-ΚB, P65, ERK and MAPK expression [62,63,64,65,66]. Discovery of TLR4/SYK axis may provide a new target for drug research on mastitis.

## 5. Conclusions

SYK is an important candidate gene for mastitis resistance traits in Chinese Holstein. The TLR4–SYK signal axis is involved in the immune cascade response of dairy cow mammary gland tissue. More importantly, the TLR4–SYK signal axis plays an important role in the immune response of bMECs with LPS or GBS. SYK and the TLR4 protein were negatively correlated in BMECs under LPS or GBS stimulation. SYK can regulate the expression of *IL-1β*, *IL-8*, *NF-κB* and *NLRP3* to inhibit the inflammatory response of bMECs. It is possible that the potential SYK–AKT1 signal axis participated in the immune response of bMECs. In conclusion, our research data supported the presence of the TLR4–SYK signal axis in bMECs to regulate immune and inflammatory response to LPS or GBS.

## Figures and Tables

**Figure 1 biomedicines-11-00097-f001:**
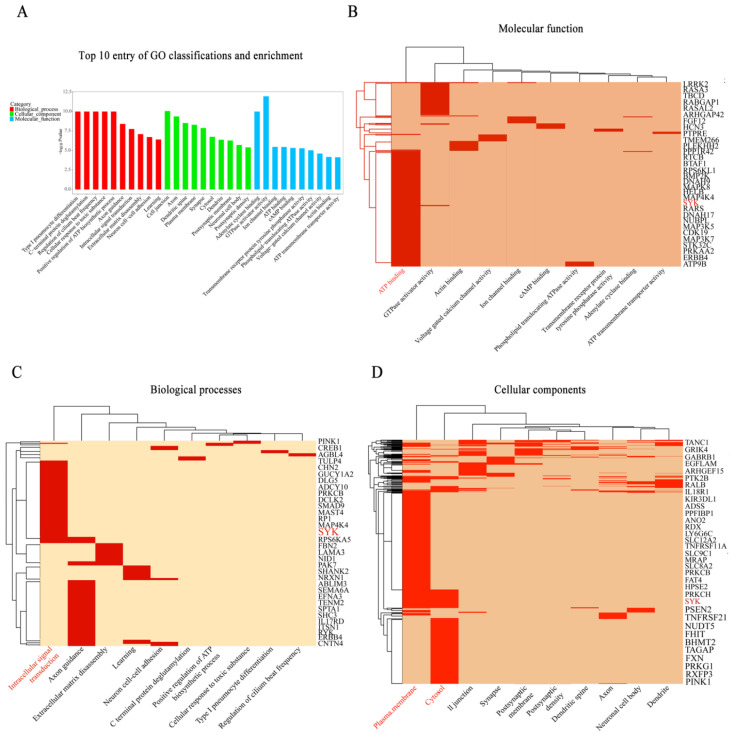
Gene functional annotation and enrichment analysis for bovine mastitis (20 normal cows and 20 mastitis cows). (**A**) The top 10 entry of three GO classifications and enrichment (molecular function, biological processes and cellular components); (**B**) gene list of the molecular function: SYK in the ATP binding entry (red marker); (**C**) gene list of the biological processes: SYK in the plasma membrane and cytosol entry (red marker); (**D**) gene list of the biological processes: SYK in the plasma membrane and cytosol entry (red marker).

**Figure 2 biomedicines-11-00097-f002:**
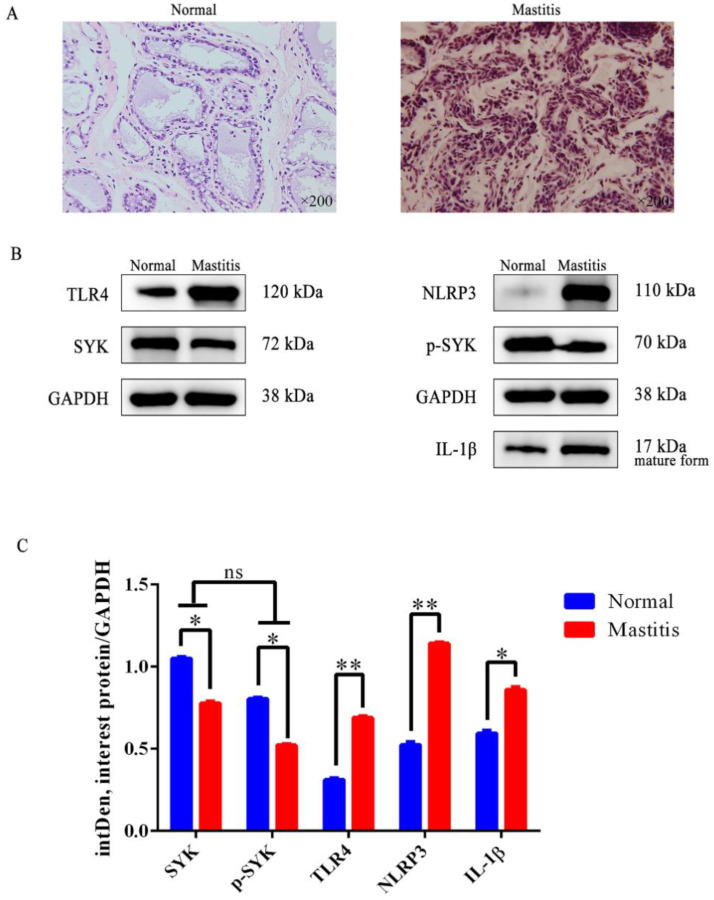
TLR4, SYK, p-SYK, IL-1β and NLRP3 protein levels in bovine normal mammary gland and mastitis tissues (3 normal cows and 3 mastitis cows). (**A**) Morphological differences in normal bovine mammary gland and mastitis tissue (H&E staining); (**B**) TLR4, SYK, p-SYK, IL-1β and NLRP3 levels were detected in bovine normal mammary gland and mastitis tissues via the WB method; (**C**) the relative quantitative method of intDen value to calculate the difference of TLR4, SYK, p-SYK, IL-1β and NLRP3 levels in bovine mammary gland and mastitis tissues. * *p* < 0.05, ** *p* < 0.01, ns = no difference.

**Figure 3 biomedicines-11-00097-f003:**
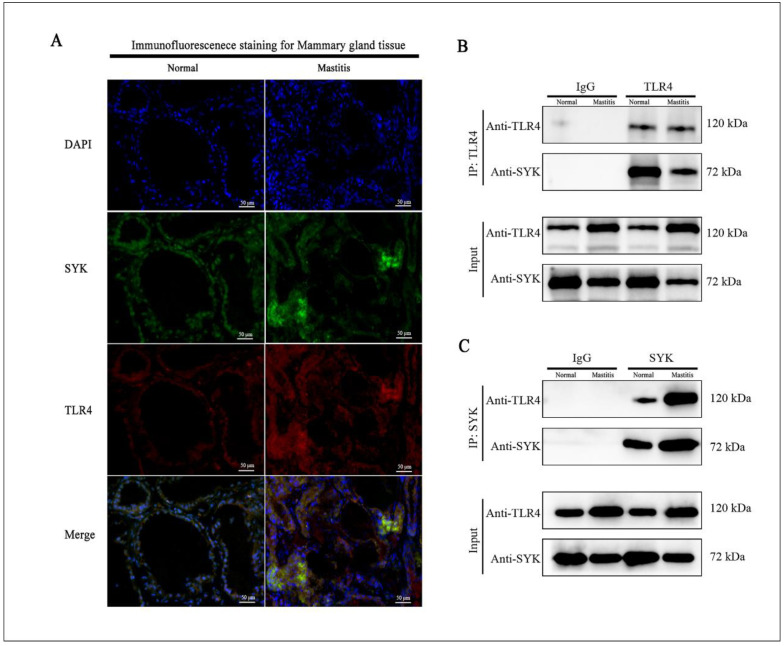
The correlations between TLR4 and SYK protein in bovine normal mammary gland and mastitis tissues (3 normal cows and 3 mastitis cows). (**A**) Immunofluorescence colocalization of TLR4 and SYK in bovine normal mammary gland and mastitis tissues, TLR4 and SYK colocalize in the mammary epithelium (normal: low-to-moderate correlation, PCC = 0.5187; mastitis: high correlation, PCC = 0.8857); (**B**,**C**) coimmunoprecipitation of TLR4 and SYK in bovine normal mammary gland and mastitis tissues (B: TLR4 pulled SYK; C: SYK pulled TLR4).

**Figure 4 biomedicines-11-00097-f004:**
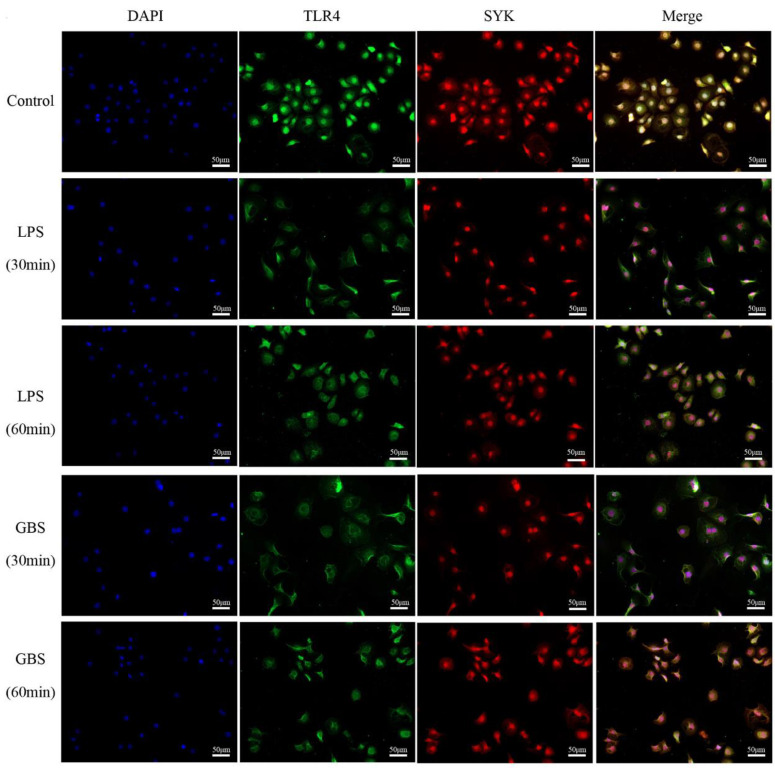
Immunofluorescence colocalization analyses of TLR4 and SYK in bMECs with LPS stimulation or GBS infection.

**Figure 5 biomedicines-11-00097-f005:**
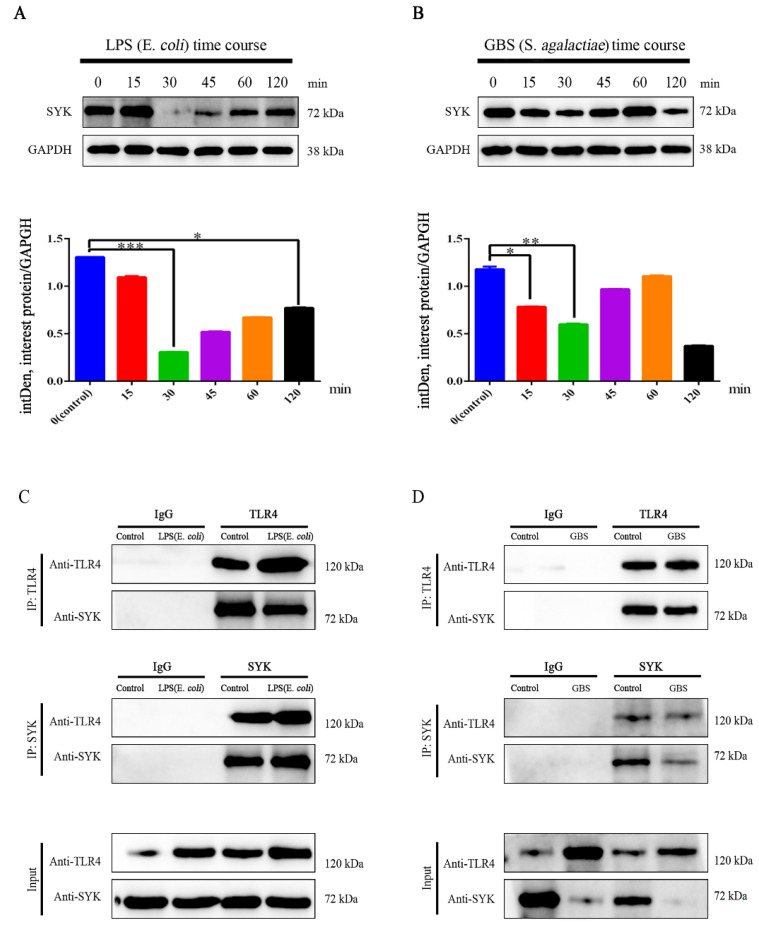
SYK levels, correlation between TLR4 and SYK in bMECs with LPS or GBS 30 min. (**A**,**B**) SYK protein levels in bMECs with LPS (*E. coli*, 25 μg/mL) stimulation or GBS (MOI = 10) infection (time course); (**C**) coimmunoprecipitation of TLR4 and SYK in bMECs with LPS (Up: TLR4 pulled SYK; Middle: SYK pulled TLR4; Down: input); (**D**) coimmunoprecipitation of TLR4 and SYK in bMECs with GBS (Up: TLR4 pulled SYK; Middle: SYK pulled TLR4; Down: input). * *p* < 0.05, ** *p* < 0.01, *** *p* < 0.001.

**Figure 6 biomedicines-11-00097-f006:**
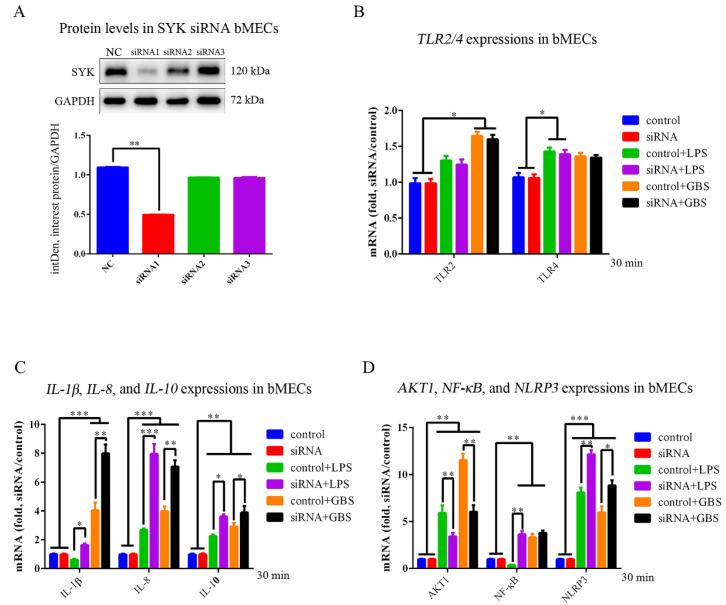
*SYK* gene siRNA interference and association genes expression in bMECs with LPS or GBS. (**A**) WB detection band showed that siRNA1 had the best interference effect in bMECs^SYK-^ and for subsequent experiments; (**B**) *TLR2* and *TLR4* gene expressions in bMECs and bMECs^SYK-^ with LPS and GBS; (**C**) *IL-1β*, *IL-8* and *TLR4* gene expressions in bMECs and bMECs^SYK-^ with LPS and GBS; (**D**) *AKT1*, *NF-κB* and *NLRP3* gene expressions in bMECs and bMECs^SYK-^ with LPS and GBS. * *p* < 0.05, ** *p* < 0.01, *** *p* < 0.001.

**Figure 7 biomedicines-11-00097-f007:**
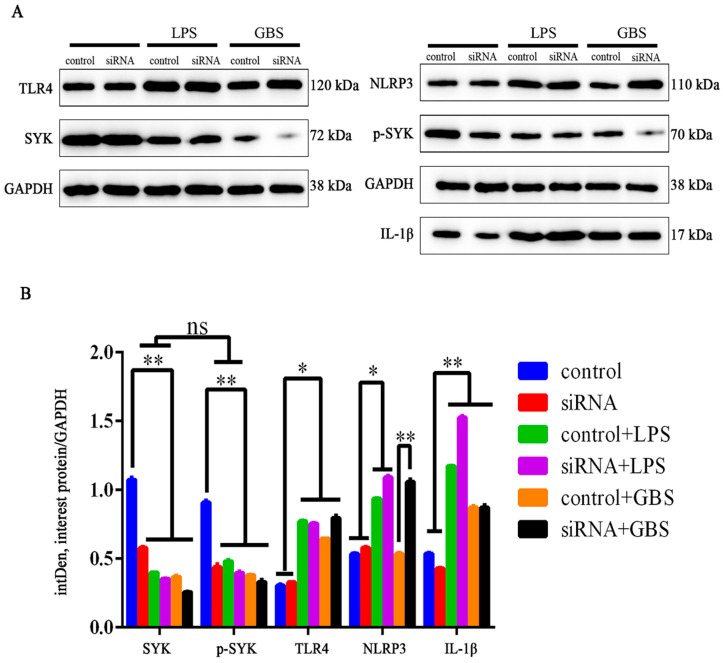
SYK, TLR4, IL-1 and NLRP3 protein levels in bMECs and bMECs^SYK-^ with LPS or GBS (30 min). (**A**) TLR4, SYK, p-SYK, IL-1β and NLRP3 protein levels were detected in bMECs and bMECs^SYK-^ with LPS or GBS via the WB method; (**B**) the relative quantitative method of intDen value to calculate the difference of TLR4, SYK, p-SYK, IL-1β and NLRP3 protein levels in bMECs and bMECs^SYK-^. * *p* < 0.05, ** *p* < 0.01, ns = no difference.

**Figure 8 biomedicines-11-00097-f008:**
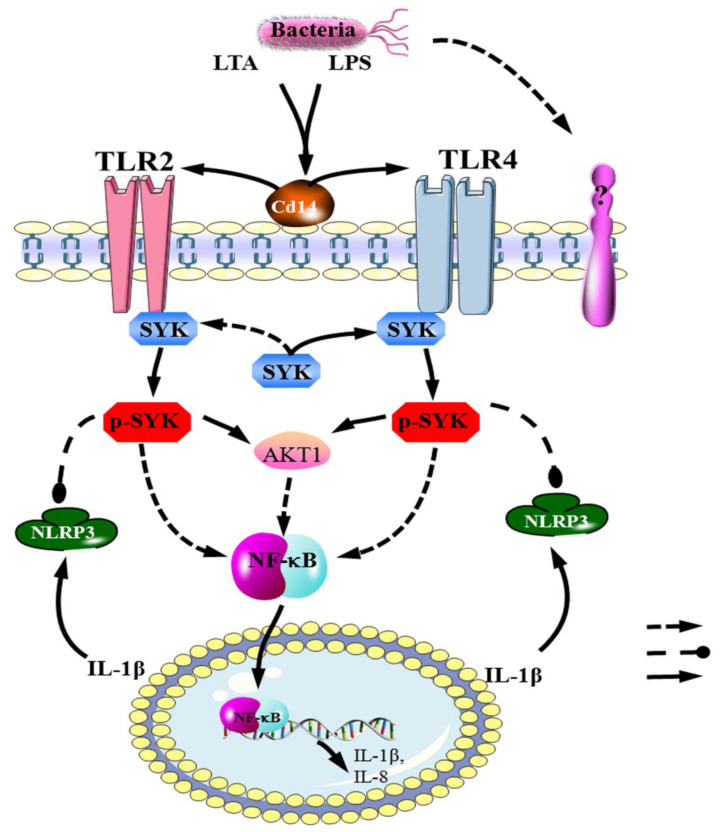
The TLR4–SYK signal axis exists in bMECs with LPS or GBS.

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
