# Peer review of "A Novel TLR4-SYK Interaction Axis Plays an Essential Role in the Innate Immunity Response in Bovine Mammary Epithelial Cells"

_biomedicines, 2022, doi:10.3390/biomedicines11010097_

Round 1

Reviewer 1 Report

Introduction

This section is complicated as it is. Possibly, the authors with to divide it into two subsections.

Also, the objectives of this work are not mentioned at all. These must be described clearly.

M – M

What exactly is acinar tissue (wrong terminology), must be corrected.

Please describe the details of the biopsy procedure.

Description of PCRs: all such descriptions should NOT be limited to presenting the primers only; other conditions (product size, annealing temperature etc.) must also be presented; the relevant references must be cited. Also such tables to be transferred to supplementary material.

The descriptions in this section are very bad and must be written in correct English.

Results

Figure 2. Please present descriptions for the histological figures.

Discussion

Please add a new subsection with the clinical significance of this work.

Reviewer 2 Report

In this paper, the authors validated the results obtained in a previous work in which a connection between TLR4 and SYK gene was postulated. A big effort was made by means of different experiments to obtain interesting results that give substance to the hypothesis. The presentation of the methodology is insufficiently detailed and has to be improved for clarity. Even the results need a better presentation to the reader to highlight their novelty as well the discussion session. Specific comments are below reported:

Line 106: the tissue description for the healthy group cow the description is : swollen, fever, and paintful?. Any description for the clinical mastitis group?

Line 115-123: This paragraph must be rephrased with a better English description. Any RIN control for RNA? For RT-qPCR experiment, the MIQE guideline has to be followed even in the description of the experiment (Bustin et al. Clinical Chemistry 55:4, 611–622 (2009),  Bustin et al. BMC Molecular Biology 2010, 11:74). Explain why only actin gene was chosen

Line 125-134: insert the ref for RIPA and BCA method. Include a brief description for WB methods

Line 123: in the supplementary file some errors in the presentation are present. Move the legend of the figure 2 in the appropriate position

Line 136-141 add “in milk” in the paragraph title;  detail the bacteria for which the selective medium is used and a brief description of the procedure. When the milk sample was taken for the isolation of pathogenic microorganisms? The milking was handmade? On which teat? It is important to find clear correspondence between the analyses conducted on milk and mammary gland

Line 150-151: detail here the specifics of the primers used in the PCR. What do you want to amplify? Which is the information reported in the bracket, to what is relative?

Line 154-155: specify the software used for the alignment  and database

Line 156: add a reference to the healthy group from which bMEC are derived

Line 176: include somewhere the words “in vitro” in the paragraph’s title

Line 187: briefly explain what MOI does mean. Not clear the connection of this experiment with the term GBS named in the paper

Line 189: any RIN control for RNA? For RT-qPCR experiment, the MIQE guideline has to be followed even in the description of the experiment (Bustin et al. Clinical Chemistry 55:4, 611–622 (2009),  Bustin et al. BMC Molecular Biology 2010, 11:74). Why RNA extraction was conducted with two different methods from the same bMEC in experiment with LPS and pathogenic microorganism

Line 192: this is a result or it has to be rephrased. A better description on the flow chart of the experiment has to be added

Line 199-210: not present the staining of bMECs. Explain somewhere the DAPI control

Line 211-218: collapse this part with 124-135 and describe the methods for WP and Co-immunoprecipitation

Line 227: RESULTS not clear the connection between the described results and some part  in methods. What is the link of the milk analysis with the results? The flowchart beside the experiment has to be clarified

Line 228-242: all this part is relative to previous work and is not a result. It has to be stated in the introduction and the relative results of ref.40 have not to be included in this work. (fig. in the paper and supplementary file)

Line 256: Supplementary fig. 4 should be included in the paper

Line 260: description of the test to detect SYK fosforilation has to be included

Line 275: The title should be Co-localization and co-immunoprecipitation instead of Interactions

Line 280-282: the Pearson’s correlation coefficient should be reported as the r value considering that you are doing a comparison of 2 protein values  and the p value in whatever experiment. Should the correlation between SYK and THL4 be negative in the mastitic group considering the results obtained in the stimulation od bMEC with LPS or GBS?

Line 286: it is the first time that you mention GBS while you have to introduce it in the materials and methods Explain better the legend of the different panels of the  supplementary Fig. 5

Line 286: explain better the Fig.3 B and C panels because you did not describe anywhere the WB experiment.

Line 322: detailed description of the supplementary fig.6 is needed. Not clear because this figure is in the supplementary and Fig 6 panel B, C and D in the paper.

Line 327-336: the description of these results is complicated by the multiple comparisons. Try to be more linear in what you want to highlight

Line 361: change C with B

Line 371: reference 42-44 are more general and not appropriate in the section

Line 371: it is not this study

Line 412: the phrase is misleading and too generic

Line 366-456: the discussion section has to be conducted in a  more appropriate way. Some parts are too general and some statements have to be included in the introduction. The paper has to report the novelty that this work gives relative to the wide bibliography of the argument. Considering the finding of the research more emphasis should be given to the explanation of figure 8

Line 451: Fig.8 can not be in the conclusion but it has to be included in the discussion section

Round 2

Reviewer 1 Report

The changes indicated by the authors were NOT made in the revised manuscript...............

This is totally unacceptable.

Reviewer 2 Report

Supplementary files legends have to be included somewhere with a description

Response 3

The author included the reference to MIQE guidelines in the text  but did not follow the procedures and did not give any procedure description in  the supplementary file 1

Response 4

The references have not been included in the text and there is a DOI to be erased

“Protein levels of SYK and related proteins were analyzed by conventional WB techniques” has to be moved before the method description

Response 5

The supplementary file uploaded in a previous version were different from the actuals and the reviewer comments were relative to those

Response 6

Delete the double phrase Inverted the culture dish, and placed in 37 °C incubator for 24 hours to obtain single colony clone of the pathogens

Please include the pathogen microorganism for which the selective medium are included

Response 7 - Response 8

Again the part with amplification is not well described. The position of the primers in the bracket has to be relative to a reference sequence and the flow chart of the sequencing and alignment with the NCBI database has to be clarified

Response 11

What do you explain to the reviewer has to be added in the paragraph to not be too generic

Response 13

The phrase “Building three pairs of siRNA sequences induced post-transcriptional silencing of the 223

SYK gene in bMECs” is a result and goes in the appropriate section.

Response 14

You have to include in the text

Response 17

You cannot include results from a previous work in this paper’s results. They should go in the introduction to justify the proceeding of the work

Response 21

Any suggested correction was done
